# Antimicrobial Resistance Profile of Bacterial Isolates from Urinary Tract Infections in Companion Animals in Central Italy

**DOI:** 10.3390/antibiotics11101363

**Published:** 2022-10-06

**Authors:** Camilla Smoglica, Giulia Evangelisti, Caterina Fani, Fulvio Marsilio, Michele Trotta, Francesca Messina, Cristina Esmeralda Di Francesco

**Affiliations:** 1Post-Graduation School of Animal Health, Breeding and Zootechnical Productions, Faculty of Veterinary Medicine, University of Teramo, Loc. Piano D’Accio, 64100 Teramo, Italy; 2CDVet Laboratorio Analisi Veterinarie, 00172 Rome, Italy

**Keywords:** antimicrobial resistance, companion animals, urinary tract infections

## Abstract

The available data on antimicrobial resistance in pets are limited compared to those collected for food-producing animals. Bacterial urinary tract infections are some of the most important indications for antimicrobial use in pets, and empiric antimicrobial treatments are often administered in the presence of clinical signs. In this study, the results obtained from the laboratory investigations carried out on dogs and cats with urinary tract infections coming from veterinary clinics and practices in Central Italy were evaluated to provide additional data concerning the bacterial urinary pathogens and their antimicrobial resistance patterns in pets. A total of 635 isolates were collected from urine samples. *Escherichia coli* was the most common species recovered in dogs and cats, followed by *Proteus mirabilis* and *Enterococcus* spp. Furthermore, it was possible to isolate bacteria not usually described in other studies concerning pets such as *Pantoea dispersa*, *Raoultella ornithinolytica*, and *Pasteurella pneumotropica* (also known as *Rodentibacter pneumotropicus*). Based on the antimicrobial susceptibility results, 472/635 (74.3%) isolates were resistant to at least one antibiotic and 285/635 (44.8%) isolates were classified as multidrug-resistant. Monitoring the antibiotic resistance profiles in pet infections is important not only for the public health implications, but also to collect data useful for the treatment of diseases in pets.

## 1. Introduction

Since their discovery, antibiotics have represented a powerful tool for the treatment of bacterial infections, with high incidence and significant lethality rates for humans and various animal species. The consumption of these drugs has continued to increase in the 21st century, thanks to the improved access to antibiotics by developing countries. Between 2000 and 2015, there was a 65% increase in the global consumption of antibiotics, which was inversely correlated with a decrease in deaths from infectious diseases [1].

Unfortunately, bacteria and other pathogens have always evolved in ways that can permit themselves to resist to the new drugs used in therapy. Indeed, antimicrobial resistance is defined as the ability of a microorganism to resist the effect of a normally active concentration of an antimicrobial agent. The result of this adaptive evolution has become a general problem and a serious threat to the treatment of infectious diseases in both human and veterinary medicine [2,3].

Many countries have started surveillance programs and are developing increasingly stringent decisions regarding the use of antibiotics in animals (mainly food-producing animals) [4]. Considering the European context, the legislation will be ever more stringent about the use of antibiotics in veterinary settings, and the application of these drugs will be regulated by the concept of rational and prudent use by means of a tailored approach to the patient and pathology [5,6,7]. This approach increasingly requires veterinarians to use diagnostic procedures that include microbiological testing for antibiotic use [6,8].

Furthermore, the emergence of multidrug-resistant bacteria (MDR) (isolates resistant to three or more antimicrobial categories) in companion animals is an increasing concern and it creates questions about the role of companion animals as potential reservoirs of resistant bacteria [9,10,11,12].

The available data for pets are limited compared to those collected for food-producing animals [4]. Most of the available information in the scientific literature is focused on the relationships existing among bacteria detected in companion animals and humans [12,13,14]. Pet-associated zoonoses are usually sporadic, and it is complicated to recognize the disease route of transmission (animal to human or vice versa) [12]. Therefore, the role of dogs and cats as potential sources of various zoonotic bacteria may not be ruled out considering that the transmission may occur via the oral–fecal route, wounds (e.g., dog bites and cat scratches), vectors (e.g., ticks), or environmental contamination [12,15,16].

An overview of antibiotic resistance profiles in pet infections is important not only due to the public health risk, but also in collecting data useful for the treatment of diseases in pets. This information may be functionally used to increase pet welfare, to decrease the risk of antibiotic resistance related to the misuse of antibiotics, and to reduce the economic cost of therapy by using short and tailored treatments [6,8,17].

Bacterial urinary tract infections (UTIs) are some of the most important indications for antimicrobial use in veterinary medicine, and they contribute to the development of antimicrobial resistance [18,19]. Indeed, empiric antimicrobial treatments are often administrated in the presence of clinical signs of UTIs, as lower urinary tract symptoms (e.g., pollakiuria, strangury, hematuria, or a combination of these signs), along with urine cytological findings, are eventually followed by the urine culture test [19,20].

Cystitis and lower urinary tract infections are generally more common in dogs than in cats [18]. The highest frequency of affected animals is described in females and neutered males [17,21], and the most common bacteria related to UTIs are *Escherichia coli* (estimated at 70–75% of cases), followed by Staphylococci, *Proteus* spp., and Enterococci [22,23]. A similar distribution was also found in human medicine, where the most common causal agent remains the uropathogenic *Escherichia coli* (UPEC) [24,25,26]. 

Therefore, systematic surveillance activity covering the most frequent bacterial species responsible for UTIs in pets and their antimicrobial resistance profiles is often lacking in veterinary clinics and hospitals, making correct antibiotic stewardship more difficult [4].

In the present study, the results obtained by laboratory investigations carried out on dogs and cats with UTIs coming from veterinary clinics and practices in Central Italy are reported in order to provide additional data concerning the frequency of bacterial urinary pathogens and their antimicrobial resistance patterns in pets.

## 2. Results

### 2.1. Bacterial Isolates

Table 1 summarizes all recovered bacterial strains and their distributions in dogs and cats. A total of 635 isolates (540 Gram-negative and 95 Gram-positive bacteria) were collected from urine samples with a statistically significant difference between the Gram-negative and Gram-positive distributions between dogs and cats (*p =* 0.001). In detail, 406 Gram-negative and 44 Gram-positive bacteria were collected from dogs, and 134 Gram-negative and 51 Gram-positive bacteria were recovered from cats.

*Escherichia coli* was the most common species recovered from dogs and cats, followed by *Proteus mirabilis* in canine samples and *Enterococcus* spp. in feline samples. Additionally, significant differences in *Pseudomonas luteola* (*p =* 0.027), *E. coli* (*p =* 0.001), *P. mirabilis* (*p =* 0.001), *Enterococcus faecalis* (*p =* 0.001), and *Staphylococcus aureus* (*p =* 0.027) frequency were observed between dogs and cats.

### 2.2. Antimicrobial Susceptibility Tests

Based on the antimicrobial susceptibility results, 472/635 (74.3%) isolates were resistant to at least one antibiotic and 285/635 (44.8%) were classified as MDR (Appendix A). In detail, 318/450 (70.6%) isolates from dogs and 154/185 (83.2%) isolates from cats were resistant to at least one antibiotic, while the MDR strains equaled 194/450 (42.2%) in dogs and 91/185 (49.1%) in cats (Table 2). 

A significant difference in resistant bacteria (*p =* 0.001) was described between dogs and cats, while no significant difference was reported for MDR bacteria. The most isolated MDR bacterial species were *P. mirabilis* and *E. coli*, followed by *Pseudomonas aeruginosa* and *Enterococcus faecalis*. In detail, the MDR *P. mirabilis* isolates were mainly identified in dogs and MDR *E. faecalis* in cats (Table 2).

The statistical analysis showed a significant difference between dogs and cats for the *Klebsiella* spp. isolates’ resistance frequency against amoxicillin/clavulanic acid (*p =* 0.016), ceftiofur (*p =* 0.014), cefalexin (*p =* 0.014), cefpodoxime (*p =* 0.014), doxycycline (*p =* 0.01), enrofloxacin (*p =* 0.005), trimethoprim/sulfamethoxazole (*p =* 0.004), tetracycline (*p =* 0.001), cefovecin (*p =* 0.016), and pradofloxacin (*p =* 0.021). Additionally, this difference was also reported for *P. mirabilis* isolates resistant to imipenem (*p =* 0.004) and for *Enterococcus* spp. isolates resistant to amoxicillin/clavulanic acid (*p =* 0.001), nitrofurantoin (*p =* 0.001), tetracycline (*p =* 0.001), and neomycin (*p =* 0.001). 

A univariable logistic regression analysis was applied to estimate risk factors associated with resistant and multidrug-resistant bacteria (Table 3). For dog isolates, the recurrent infections were significantly associated with resistant (OR: 2.36, 95%CI:1.16–4.81) and MDR bacteria (OR: 2.54, 95%CI: 1.46–4.43). Both dogs (OR: 2.55, 95%CI:1.4–4.65) and cats (OR: 3.72, 95%CI: 1.49–9.27) treated with antibiotic therapy had a higher risk of harboring MDR bacteria in comparison to animals that did not receive antibiotics. Finally, resistant bacteria in dogs and MDR bacteria in cats were more likely detected in females and sterilized females. Other variables were not significantly associated with the resistant and MDR bacteria (*p* > 0.05).

### 2.3. Antibiotic Susceptibility Test in Canine Bacteria

The main results of the antibiotic susceptibility test in canine isolates are reported in Figure 1. In detail, the *E. coli* isolates in dogs were mostly resistant to ampicillin (125/263; 47.5%), followed by cephalothin (102/263; 38.7%), cefalexin (83/263; 31.5%), and cefpodoxime (82/263; 31.1%). All isolates were susceptible to amikacin.

All isolates of *Klebsiella* spp. were found to be resistant to ampicillin (21/21;100%) and mainly resistant to nitrofurantoin (11/21; 52.3%). All isolates were sensitive to amikacin and imipenem.

*Pseudomonas* spp. isolates are intrinsically resistant to diverse antibiotics such as cephalosporines and trimethoprim/sulfamethoxazole. Additionally, the majority of isolates also showed resistance profiles to ampicillin (17/18; 94.4%), nitrofurantoin (17/18; 94.4%), and tetracycline (16/18; 88.8%). The susceptibility was reported for amikacin in all isolates and for doxycycline, gentamicin, and imipenem in 17/18 (94.4%) isolates.

The intrinsic resistance to ampicillin, cefazolin, tetracycline, and nitrofurantoin was confirmed for *Proteus* spp. isolates. Some of them were also resistant to imipenem (51/83; 61.4%), chloramphenicol (41/83; 49.3%), trimethoprim/sulfamethoxazole (31/83; 37.3%), and enrofloxacin (19/83; 22.8%). The highest levels of susceptibility were detected for neomycin (83/83;100%) and amikacin (79/83; 95.1%).

*Enterococcus* spp. were found to be particularly resistant to cefovecin (26/47; 55.3%), erythromycin (17/47; 36.1%), and doxycycline (17/47; 36.1%), while they were susceptible to florfenicol (46/47; 97.8%), chloramphenicol (44/47; 93.6%), and nitrofurantoin (43/47; 91.4%).

Finally, *Staphylococcus* spp. isolates were mainly resistant to erythromycin (10/14; 71.4%) and benzylpenicillin (12/14; 85.7%) with susceptibility to amikacin (14/14; 100%), gentamicin (13/14; 92.8%), and neomycin (13/14; 92.8%). 

### 2.4. Antibiotic Susceptibility Test in Feline Bacteria 

The main outcomes of the antibiotic susceptibility test of feline bacteria are shown in Figure 2. In detail, *E. coli* isolates in cats were found to be mostly resistant to ampicillin (44/80; 55%), followed by cephalothin (34/80; 42.5%), cefalexin (25/80; 31.2%), and cefpodoxime (28/80; 35%). All isolates were susceptible to amikacin and imipenem.

Considering that *Pseudomonas* spp. are intrinsically resistant to diverse antibiotics such as cephalosporines and trimethoprim/sulfamethoxazole, many isolates belonging to this species also showed resistance profiles to ampicillin (14/17; 82.3%), nitrofurantoin (12/17; 70.5%), and tetracycline (16/17; 94.1%). The susceptibility was reported for amikacin in all isolates and for gentamicin and imipenem in 15/17 (88.2%) isolates.

The *Proteus* spp. isolates, which are intrinsically resistant to ampicillin, cefazolin, tetracycline and nitrofurantoin, were also found to be resistant to trimethoprim/sulfamethoxazole (5/16; 31.2%), chloramphenicol (6/16; 37.5%), and imipenem (8/16; 50%). The highest levels of susceptibility were detected for neomycin (16/16; 100%), gentamycin (16/16; 100%), and amikacin (15/16; 93.7%). 

All isolates of *Klebsiella* spp. resulted resistant to ampicillin (8/8; 100%) and enrofloxacin (8/8; 100%). In addition, these isolates showed high levels of resistant to marbofloxacin (7/8; 87.5%), tetracycline (7/8; 87.5%), cephalothin (7/8; 87.5%), ceftiofur (7/8; 87.5%), cefalexin (7/8; 87.5%), and cefpodoxime (7/8; 87.5%). All isolates were susceptible to amikacin, gentamicin, imipenem, and neomycin.

The *Enterococcus* spp. isolates were particularly resistant to cefovecin (38/40; 95%), marbofloxacin (21/40; 52.5%), and doxycycline (17/40; 42.5%), while they were susceptible to amoxicillin/clavulanic acid (36/40; 90%), benzylpenicillin (36/40; 90%) and nitrofurantoin (39/40; 97.5%).

Finally, the *Staphylococcus* spp. isolates were found to be mainly resistant to benzylpenicillin (9/11; 81.8%), erythromycin (7/11; 63.6%), enrofloxacin (6/11; 54.5%), marbofloxacin (6/11; 54.5%), and pradofloxacin (6/11; 54.5%) with susceptibility to amikacin (11/11; 100%), gentamicin (9/11; 81.8%), doxycycline (10/11; 90.9%), and neomycin (10/11; 90.9%).

## 3. Discussion

The aim of this study was to increase the knowledge on the antimicrobial resistance of pathogens associated with UTIs in dogs and cats from Central Italy. This study is an update of the available data considering that the last similar study was performed on isolates collected during 2013–2015 in Central Italy [27]. In the present study, a higher number of bacteria was collected and a full panel of antibiotics was applied for the antibiotic susceptibility test compared to the previous study [27]. Additionally, in our study, the highest frequency of Gram-negative isolates was obtained. The percentages of resistance to antibiotics such as amoxicillin/clavulanic acid, cephalexin, doxycycline, gentamicin, and marbofloxacin observed in *E. coli*, *Pseudomonas* spp., *Klebsiella* spp., *Proteus* spp. and *Enterobacter* spp. were lower in both canine and feline bacteria compared to the previous investigation.

The risk factors for bacterial UTIs were previously investigated considering their age, breed, sex, urethral catheterization, hospital admission, year of sample collection, and concurrent disease (as neurologic pathology) [25,26]. In the present study, age, breed, sex, antibiotic therapy, concurrent diseases, and recurrent and chronic infections were assessed as potentially related determinants for resistant and MDR bacteria responsible for UTIs in dogs and cats. In particular, resistant bacteria in dogs and MDR bacteria in cats were more likely detected in females and sterilized females, as previously described by Thompson et al. [21] and Fonseca et al. [17]. In addition, the antibiotic treatments were linked to the occurrence of MDR bacteria in dogs and cats, confirming what previously reported from other studies in Brazil [25], Portugal [28], and France [29]. Finally, the recurrent diseases were significantly associated with resistant and MDR bacteria in dogs. A previous study also described as a risk factor of resistant bacteria in UTIs the chronic infections evaluated here, along with recurrent infections [25]. The choice to consider chronic and recurrent diseases as different variables in the present study may have affected this result, suggesting that the recurrent UTIs are associated with repeated therapies over time.

Overall, a predominance of Gram-negative bacteria over Gram-positive isolates was observed, even if a significant difference in distribution emerged between dogs and cats. Indeed, the highest frequency of Gram-positive isolates was detected in feline patients, as also described in a recent study by Fonseca et al. [17].

*Escherichia coli* is the most common bacteria responsible for UTIs in dogs and cats in Europe [22,23], the USA [19], Canada [30], and New Zealand [31]. The same result was reported in the most recent studies carried out in the USA [32,33], the UK [17], Spain [34], Italy [27], and Germany [35]. Cephalosporins resistance above 30% was detected in *E. coli* isolated from dogs and cats. In detail, the isolates were resistant to first-generation (cephalothin, cephalexin) or third-generation cephalosporin (cefpodoxime). These data agree with the abovementioned reports highlighting the need for the continuous monitoring of cephalosporins resistance and to enhance the antibiotic stewardship for these molecules [17,27].

*Proteus mirabilis* was the second most common *Enteobacteriales* isolated in dogs, as previously reported by other authors [17,23,36]. This species was widely investigated in different studies performed in Spain [34], the UK [17], Thailand [26], and the USA [33], or in retrospective studies of data collected in Europe [22,23]. This bacterium is of particular interest considering its intrinsic resistance to different antibiotics and its acquired fluoroquinolone resistance, as described several times in the abovementioned studies. In the present study, the resistant profile observed for *P. mirabilis* was also characterized by the resistance to imipenem, as recently reported in the USA [33]. These data raise concerns regarding the potential zoonotic role of UTI-related *P. mirabilis* strains isolated in the present study. Indeed, recent investigations described *P. mirabilis* strains recovered from pets as closely related to human strains [12,13].

Other Gram-negative bacteria have increased their antibiotic resistance profiles, becoming a significant concern for public health in human and veterinary medicine [37]. In this regard, *Klebsiella* spp. and *Pseudomonas* spp. revealed high resistance to ampicillin, cephalosporins, tetracycline, and fluoroquinolones, and similar patterns were described in Portugal [12], Spain [34], Thailand [26], and the USA [33].

As previously described by other authors [17,22,34], *Enterococcus* spp. was the second most frequent urinary pathogen in feline samples. The isolates of this study resulted resistant to doxycycline and fluroquinolones, as reported in other studies carried out in Italy [27], Portugal [38], the UK [17], Spain [34], and the USA [33]. Considering the intrinsic resistance of certain *Enterococcus* spp. (i.e., *E. faecium* and *E. faecalis*) [33], the acquired resistance to tetracycline and fluoroquinolones may reduce the chance of treatments for Enterococci-related infections. Indeed, in humans, these opportunistic pathogens cause a wide range of difficult-to-treat nosocomial infections [17,37].

*Staphylococcus* spp. isolates were detected in both dogs and cats in this study. In detail, *S. pseudintermedius* was the most isolated species, followed by *S. lentus* and *S. aureus*. Since 2006, *S. aureus* has represented a significant health problem in companion animals and a public health concern [37]. However, *S. aureus* strains isolated from companion animals were mainly related to different human-associated strains, while recent data about *S. pseudintermedius* suggested the role of animals as potential reservoirs and that zoonotic transmission is currently considered plausible [39,40,41]. Oxacillin resistance was identified in 12/14 Staphylococci isolates (85.7%), which also showed resistance to erythromycin, benzylpenicillin, and fluroquinolones. Similar profiles of resistance were described in a retrospective analysis performed in Europe during the period of 2008–2013 [22,23] and in recent studies carried out in Spain [34], the UK [17], and Central Italy [27]. On the other hand, the isolates in these studies showed the highest levels of susceptibility to gentamicin, amikacin, and neomycin, as compared to recent studies performed in Portugal [38], Northwest Italy [6], Thailand [26], and the USA [33]. As previously suggested by other authors, these differences in resistance profiles may be related to the geographical and time-associated variations in antimicrobial use practices [27,33]. In particular, the study carried out in Northwest Italy was performed in a veterinary teaching hospital, a reference center for complex clinical cases [6]. Therefore, it is reasonable to assume that the patients came from private clinics where they may have received antibiotic treatments with the potential selection of resistant strains [27,32]. Based on these considerations, the knowledge on the geographical distribution of the most common UTI pathogens and their antibiotic susceptibility is necessary to ensure adequate antibiotic therapy and to preserve the antibiotic efficacy [17].

Furthermore, in the present work it was possible to isolate bacteria not usually described in other studies of UTIs in pets, such as *Pantoea dispersa*, *Raoultella ornithinolytica,* and *Pasteurella pneumotropica* (also known as *Rodentibacter pneumotropicus*). In detail, two *Pantoea dispersa* isolates were obtained from urine samples of one dog and one cat. The feline isolate was found to be susceptible to all tested antibiotics, while the canine strain was resistant to tetracycline and trimethoprim/sulfamethoxazole. This bacterium belongs to the genus *Pantoea*, which is included in the *Erwiniaceae* family, and it was frequently found in plants, soil, and water. Some *P. dispersa* isolates were related to nosocomial blood stream infections, rhinosinusitis, and hepatitis in human patients [42,43,44]. However, conventional analyses are considered ineffective in achieving the correct identification of this bacterium, resulting in the misdiagnosis and underestimation of *P. dispersa*-related infections [42,43,44].

The multidrug-resistant *Raoultella ornithinolytica* was recovered from canine urine samples in the present study. This bacterium was previously described in retail vegetables and meat [45,46], chicken products, poultry flock environments [47,48], and a septicemic calf [49]. Recently, it was isolated from feline urine in Austria, and it was evaluated as being MDR [50].

Finally, *Pasteurella pneumotropica* (also known as *Rodentibacter pneumotropicus*) was isolated in 2 dogs, showing susceptibility to all tested antibiotics. This bacterium has been previously described in the oral flora of dogs and cats [51] and has been related to infections in puppies and humans as a result of dog bites [52,53,54]. 

All of the unconventional isolates detected in this study were considered emerging Gram-negative bacteria and represent a major concern in view of their potential role as pathogens, including for humans.

Overall, gentamicin and amikacin had the highest activity levels both in canine and feline isolates in this study. As previously suggested by other authors, these results may be related to the careful dosing of this antibiotic in veterinary settings because of its toxicity [27,33]. More recent studies indicate that their use at different doses and with short duration therapies, as commonly applied in human medicine, may be a possible alternative approach for treating animals while preserving the efficacy of these antibiotics. However, more specific investigations are needed [8]. 

The retrospective evaluation of bacteria isolated from the urine samples of dogs and cats with UTIs and the assessment of their antibiotic resistance patterns may be useful for clinicians to choose a first-line drug to treat the patients. The International Society for Companion Animal Infectious Diseases (ISCAID) has published UTI treatment guidelines for dogs [8], suggesting that the first-line empirical drug choice should be based on the local prevalence of bacterial pathogens and their resistance profiles. Based on the available data for ISCAID, amoxicillin, amoxicillin/clavulanic acid, and trimethoprim/sulfamethoxazole are considered as the first empirical antimicrobial choices for UTI treatment, while nitrofurantoin, fluoroquinolones, and third-generation cephalosporins are only recommended if resistance to first-line antimicrobials is detected or the condition of the patient is serious [8]. 

However, it is important to consider that the European Medicines Agencies (EMA) has categorized antimicrobials according to the risk for public health and the need for their use in veterinary medicine [4]. Categories A and B include antimicrobials that are critically important in human medicine, whose use is limited in human medicine, and whose use is not authorized in the European Union (EU) in veterinary medicine for the treatment of production animals [55]. The highest-priority and most critically important antibiotics include molecules such as carbapenems, third-generation cephalosporins, and fluoroquinolones, which are considered the last resources to threat MDR infections in human medicine [56,57]. Therefore, the resistance profiles to carbapenems, third-generation cephalosporins, and fluoroquinolones described in this study, even in MDR bacteria, highlight the importance of investigations on UTI pathogens considering the important potential public health implications.

## 4. Materials and Methods

### 4.1. Selection Criteria and Data Sources

A retrospective study was carried out by collecting the urine culture results from dogs and cats provided by the private laboratory “CDVet Laboratorio Analisi Veterinarie” (Rome, Italy) from January 2020 to December 2020. The urine samples came from 54 first-opinion veterinary practices or 24 h clinics, mainly located in Central Italy. The anamnestic data, such as for the animal species, sex, age, breed, and sterilized status, as well as clinical details concerning concurrent diseases or antibiotic therapy, were registered. Additionally, a distinction between chronic (an infection that persists over time and is caused by the same etiological agent) and recurrent (an infection resolved in the past but recurring and caused by the same or a different etiological agent) infections was applied.

### 4.2. Identification of Isolated Bacteria and Susceptibility Testing 

The urine samples were aerobically incubated at 35 ± 2 °C for 18–24 h using the Chromatic^TM^Detection Medium (Liofilchem^®^, Roseto degli Abruzzi, Italy) for the morphological identification of grown colonies, and the VITEK^®^ system (Biomerieux, Marcy-l’Étoile, France) was applied for the species identification of bacteria. In order to identify the therapeutic options, the antibiotic susceptibility tests were carried out using the VITEK^®^ system (Biomerieux, Marcy-l’Étoile, France) according to the guidelines of the European Committee on Antimicrobial Susceptibility Testing (EUCAST) and the available veterinary clinical breakpoints from the Clinical and Laboratory Standards Institute (CLSI) [58,59].

Based on the bacterial strain identification, the panel of antimicrobial agents was used for antimicrobial susceptibility tests, including 26 drugs from 9 categories: aminoglycosides (amikacin, kanamycin, neomycin, and gentamicin); penicillins (ampicillin, amoxicillin/clavulanic acid, benzylpenicillin, and oxacillin); carbapenems (imipenem); first-generation cephalosporins (cephalothin, cephalexin); third-generation cephalosporins (ceftiofur, cefovecin, cefpodoxime); fluoroquinolones (enrofloxacin, pradofloxacin, and marbofloxacin); tetracyclines (doxycycline, minocycline, and tetracycline); macrolides (erythromycin); lincosamides (clindamycin); phenolics (florfenicol and chloramphenicol); nitrofurantoin and trimethoprim/sulfamethoxazole. 

### 4.3. Data Analysis and Statistical Analysis

The statistical analysis was performed using software packages in Stata [60]. The chi-square (*χ*2) or Fisher’s test was used to compare the frequency of resistant bacteria between animal species. The risk factors (sex, breed, age, concomitant diseases, recurrent or chronic infections, occurrence of antibiotic therapy) and resistant or MDR bacterial outcomes were assessed using logistic regression. The *p* values resulting from the statistical analysis were evaluated as significant when the values were less than 0.05.

## 5. Conclusions

The results of the present study provide further information on the antibiotic resistance profiles and risk factors of UTI-related pathogens in dogs and cats in Central Italy. The data collected highlight the relevant occurrence of *E. coli*, *Proteus mirabilis,* and *Enterococcus* spp., but also the possibility of identifying uncommon pathogens. The resistance profiles identified reveal a remarkable percentage of MDR bacteria. Additionally, the resistance profiles also include critically important antibiotics for human medicine and pathogens that have been previously related to infections in humans. Therefore, the monitoring of antibiotic resistance in the UTIs of pets should be continuously applied to preserve public health and to realize more effective therapies in clinical practice, tailored to different areas and based on data available in the literature.

## Figures and Tables

**Figure 1 antibiotics-11-01363-f001:**
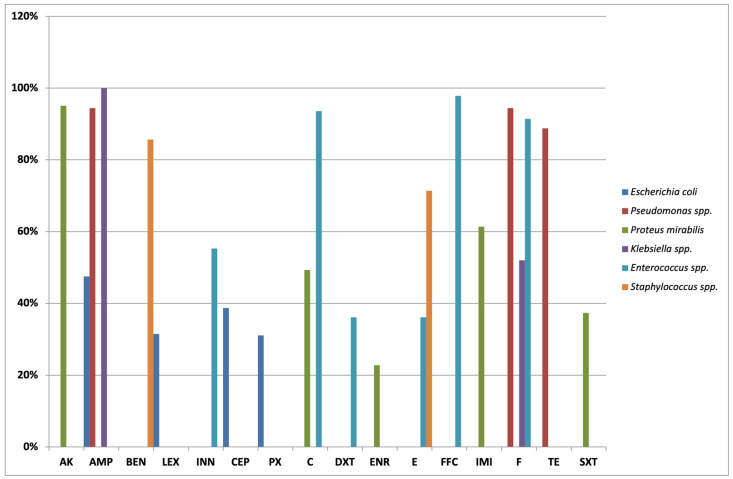
Antimicrobial resistance frequencies (% of resistant bacteria) detected in isolates from urine samples of dogs. The percentage values greater than or equal to 10% are reported. AK: amikacin; AMP: ampicillin; BEN: benzylpenicillin; LEX: cephalexin; INN: cefovecin; CEP: cephalothin; PX: cefpodoxime; C: chloramphenicol; DXT: doxycycline; ENR: enrofloxacin; E: erythromycin; FFC: florfenicol; IMI: imipenem; F: nitrofurantoin; TE: tetracycline; SXT: trimethoprim/sulfamethoxazole.

**Figure 2 antibiotics-11-01363-f002:**
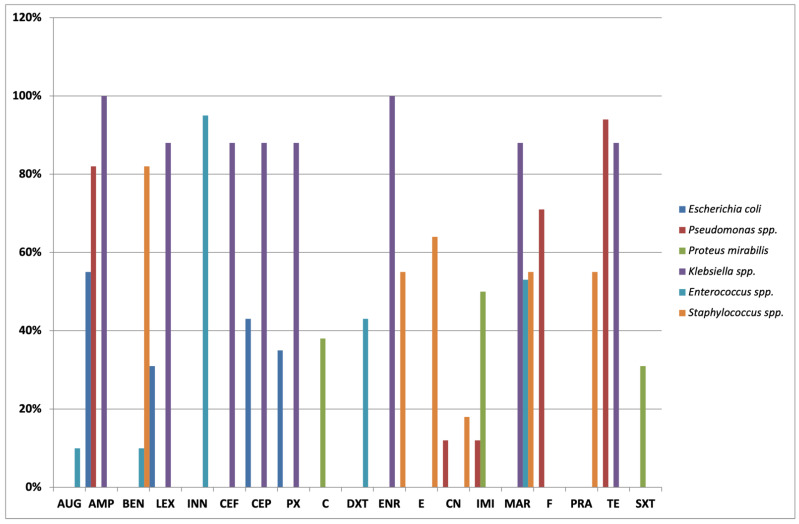
Antimicrobial resistance frequencies (% of resistant bacteria) detected in isolates from urine samples of cats. The percentage values greater than or equal to 10% are reported. AUG: amoxicillin/clavulanic acid; AMP: ampicillin; BEN: benzylpenicillin; LEX: cephalexin; INN: cefovecin; CEF: ceftiofur; CEP: cephalothin; PX: cefpodoxime; C: chloramphenicol; DXT: doxycycline; ENR: enrofloxacin; E: erythromycin; CN: gentamicin; IMI: imipenem; MAR: marbofloxacin; F: nitrofurantoin; PRA: pradofloxacin; TE: tetracycline; SXT: trimethoprim/sulfamethoxazole.

**Table 1 antibiotics-11-01363-t001:** Bacterial species isolated from urine samples of investigated dogs and cats.

Bacterial Species	% in Dogs (N = 450)	% in Cats (N = 185)	Total	*p* Value
*Acinetobacter* spp.	0.22 (N = 1)	1.08 (N = 2)	3	*p =* 0.204
*Enterobacter aerogenes*/*Klebsiella aerogenes*	0.66 (N = 3)	0	3	N/A
*Enterobacter cloacae complex*	2.21 (N = 10)	5.4 (N = 10)	20	*p =* 0.066
*Enterococcus casseliflavus*	0.22 (N = 1)	0.54 (N = 1)	2	*p =* 0.498
*Enterococcus faecalis*	3.11 (N = 14)	18.91 (N = 35)	49	*p =* 0.001
*Enterococcus faecium*	2.44 (N = 11)	1.08 (N = 2)	13	*p =* 0.365
*Enterococcus* spp.	0.88 (N = 4)	1.08 (N = 2)	6	*p =* 1
*Escherichia coli*	58.84 (N = 263)	43.24 (N = 80)	343	*p =* 0.001
*Klebsiella oxytoca*	0.22 (N = 1)	0.54 (N = 1)	2	*p =* 0.498
*Klebsiella pneumoniae*	4.44 (N = 20)	3.78 (N = 7)	27	*p =* 0.707
*Pantoea dispersa*	0.22 (N = 1)	0.54 (N = 1)	2	*p =* 0.498
*Pasteurella pneumotropica*/*Rodentibacter**pneumotropicus*	0.44 (N = 2)	0	2	N/A
*Proteus mirabilis*	19.11 (N = 86)	8.64 (N = 16)	102	*p =* 0.001
*Pseudomonas aeuriginosa*	3.55 (N = 16)	6.48 (N = 12)	28	*p =* 0.102
*Pseudomonas fluorescens*	0.22 (N = 1)	0.54 (N = 1)	2	*p =* 0.498
*Pseudomonas luteola*	0.22 (N = 1)	2.16 (N = 4)	5	*p =* 0.027
*Raoultella ornithinolytica*	0.22 (N = 1)	0	1	N/A
*Staphylococcus aureus*	0.22 (N = 1)	2.16 (N = 4)	5	*p =* 0.027
*Staphylococcus lentus*	1.11 (N = 5)	0.54 (N = 1)	6	*p =* 0.677
*Staphylococcus pseudintermedius*	1.77 (N = 8)	3.24 (N = 6)	14	*p =* 0.253
Total	450	185	635	N/A

N/A: not applicable.

**Table 2 antibiotics-11-01363-t002:** Numbers of multidrug resistance bacteria in different strains isolated from urine samples of dogs and cats in Central Italy during January to December 2020. The percentage values are reported in brackets.

MDR Bacterial Species	Dogs	Cats	Total	*p* Value
*Acinetobacter* spp.	-	1 (50%)	1	N/A
*Escherichia coli*	59 (22.4%)	18(22.5%)	77	*p =* 0.99
*Enterobacter cloacae complex*	8 (80%)	9 (90%)	17	*p =* 1
*Enterococcus casseliflavus*	-	1(100%)	1	N/A
*Enterococcus faecalis*	7(50%)	19(54.2%)	26	*p =* 0.785
*Enterococcus faecium*	9(81.8%)	1(50%)	10	*p =* 0.423
*Enterococcus* spp.	2(50%)	2(100%)	4	N/A
*Klebsiella oxytoca*	-	1(100%)	1	N/A
*Klebsiella pneumoniae*	9(45%)	7(100%)	16	N/A
*Proteus mirabilis*	73(84.8%)	10(62.5%)	83	*p =* 0.034
*Pseudomonas aeuriginosa*	15(93.7%)	12(100%)	27	*p =* 1
*Pseudomonas fluorescens*	1(100%)	1(100%)	2	N/A
*Pseudomonas luteola*	-	2(50%)	2	N/A
*Raoultella ornithinolytica*	1(100%)	-	1	N/A
*Staphylococcus aureus*	-	2(50%)	2	N/A
*Staphylococcus lentus*	4(80%)	1(100%)	5	N/A
*Staphylococcus pseudintermedius*	6(75%)	4(66.6%)	10	*p =* 1
Total	194	91	285	*p =* 0.162

N/A: not applicable.

**Table 3 antibiotics-11-01363-t003:** Univariable logistic regression analysis for variables associated with resistant and multidrug-resistant bacteria isolated from urine samples of dogs and cats in Central Italy during January–December 2020.

Variables	Dog		Cats	
	Odds Ratios (95%CI)		Odds Ratios (95%CI)	
	Resistant Bacteria	MDR Bacteria	Resistant Bacteria	MDR Bacteria
**Age**				
Puppy/kitten	Reference	Reference	Reference	Reference
Subadult	1.28 (0.73–2.24)	1.05 (0.61–1.81)	0.56 (1.17–1.82)	0.79 (0.35–1.78)
Adult	1.82 (0.98–3.38)	1.3 (0.73–2.32)	0.75 (0.22–2.56)	1 (0.44–2.34)
**Sex**				
Castreted male	Reference		Reference	Reference
Male	1.73 (0.84–3.59)	1.63 (0.76–3.47)	1.66 (0.53–5.16)	0.98 (0.44–2.17)
Sterilized female	**2.45 (1.15–5.21)**	1.94 (0.9–4.18)	0.97 (0.38–2.5)	**0.27 (0.13–0.58)**
Female	**2.28 (1.08–4.83)**	1.6 (0.74–3.44)	0.58 (1.17–1.94)	**0.21 (0.06–0.66)**
**Breed**				
No	Reference	Reference	N/A	Reference
Yes	0.93 (0.61–1.4)	1.16 (0.79–1.69)	N/A	0.59 (0.2–1.7)
**Antibiotic therapy**				
No	Reference	Reference	Reference	Reference
Yes	1.42 (0.72–2.81)	**2.55 (1.4–4.65)**	6.37 (0.83–48.8)	**3.72 (1.49–9.27)**
**Concurrent diseases**				
No	Reference	Reference	Reference	Reference
Yes	0.93 (0.32–2.66)	1.19 (0.45–3.15)	3.19 (0.72–14.1)	1.64 (0.38–7.1)
**Recurrent infections**				
No	Reference	Reference	Reference	Reference
Yes	**2.36 (1.16–4.81)**	**2.54 (1.46–4.43)**	0.44 (0.15–1.26)	1.43 (0.57–3.58)
**Chronic infections**				
No	Reference	Reference	Reference	Reference
Yes	1.34 (0.67–2.66)	1.49 (0.82–2.69)	1.68 (0.36–7.71)	1.7 (0.63–4.62)

Note: 95%CI: 95% confidence interval; N/A: not applicable; bold values indicate *p* < 0.05.

## Data Availability

Not applicable.

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
