# Peer review of "Antimicrobial Resistance Profile of Bacterial Isolates from Urinary Tract Infections in Companion Animals in Central Italy"

_antibiotics, 2022, doi:10.3390/antibiotics11101363_

Round 1

Reviewer 1 Report

Your work is very relevant nowadays. It is important to do more studies about antimicrobial resistance in pets in order to compare with other countries and to stablish guidelines. 

I just have a few comments about this work.

1) The results available in the section 2.3 can be represent in a graph. I think that will be more easy to see the results. 

2) In the Discussion, in the lines 217, 228 and 230, after "by" please put the name of the author.

Author Response

Dear Reviewer 1,

We are very happy to have received a positive evaluation, and we would like to express our appreciation for the thoughtful comments and helpful suggestions.

Our detailed, point-by-point responses to the reviewer comments are given below, whereas the corresponding revisions are marked in colored text in the manuscript file.

Comment 1: The results available in the section 2.3 can be represent in a graph. I think that will be more easy to see the results. 

Figure 1 has been added to section 2.3 and Figure 2 to section 2.4 to provide graphics to aid understanding of the text.

Comment 2: In the Discussion, in the lines 217, 228 and 230, after "by" please put the name of the author.                    

The manuscript was modified as suggested by Reviewer. In detail, the name of the author was added when only one reference is reported.

Reviewer 2 Report

In their study, Smoglica and colleagues reported the data concerning the frequency of bacterial urinary pathogens and their antimicrobial resistance patterns in pets. The subject is very important because few studies are done on pets compared to those done on food-producing animals. This fact makes the manuscript potentially interesting.

In general, the manuscript seems to me an appropriate paper to be published. The introduction is appropriate and summarizes the objective of the study, the analysis methods are adequate, and the discussion responded to the research problems.

However, I would like to make some clarifications that would improve the quality of the article:

·       In Table 1, a column with p-values should be inserted to make the data clearer

·       In Table 2, the percentages of each resistant bacterium out of the total of those isolated for each species should be listed, and p-values.

·       Subpoints 2.3 and 2.4 from the Results repeat information from the supplementary material S1-S2. The percentages can be entered in those tables.

·       Some clear Conclusions should be inserted at the end.

·       Bacterial names must first appear in full, then abbreviated and always italic.

Author Response

Dear Reviewer 2,

We are very happy to have received a positive evaluation, and we would like to express our appreciation for the thoughtful comments and helpful suggestions.

Our detailed, point-by-point responses to the reviewer comments are given below, whereas the corresponding revisions are marked in colored text in the manuscript file.

Comment 1: In Table 1, a column with p-values should be inserted to make the data clearer

 The p value was added in a new column. In addition, another column with the total number of isolates was added in accordance with the Reviewer 3 suggestions.

Comment 2: In Table 2, the percentages of each resistant bacterium out of the total of those isolated for each species should be listed, and p-values.

The p value was added in a new column and the percentage of Multidrug resistance bacteria of each strains isolated was added.

Comment 3: Subpoints 2.3 and 2.4 from the Results repeat information from the supplementary material S1-S2. The percentages can be entered in those tables.

Done

Comment 4: Some clear Conclusions should be inserted at the end.

Done

Comment 5: Bacterial names must first appear in full, then abbreviated and always italic.

Done

Reviewer 3 Report

Editors of Antibiotics

Authors of the manuscript

The manuscript entitled „Antimicrobial resistance profile of bacterial isolates from urinary tract infections in companion animals in Central Italy”, presents the results of the experiment evaluating resistance of bacteria isolated from UTIs in cats and dogs in a given region. While the concept of the work and the results are of value, because therapy tailored to the local resistance profiles of microorganisms would be both beneficial to patients and prevent the build-up of resistance among bacteria, unfortunately the execution of the manuscript reduces the quality of the whole work. Together the language (English) errors and, at some points, lack of attention to the detail or use of “mental shortcuts” result in a situation where the reader sometimes has to figure out what was the exact meaning of the sentence. I would suggest English editing of the text. Please find some of my comments and suggestions in detail within the attached file. Except for a few, language errors are not indicated.

Sincerely,

Reviewer

Author Response

Reviewer 3 raised several concerns, which we have carefully considered and made every effort to address. We fundamentally agree with all the comments made by the Reviewer, and we have incorporated corresponding revisions into the manuscript (antibiotics-1945951_R1). We would like to express our appreciation to Reviewer 3 for the thoughtful comments and helpful suggestions.